# Platinum(IV) Complexes of the 1,3,5-Triamino Analogue of the Biomolecule Cis-Inositol Designed as Innovative Antineoplastic Drug Candidates

**DOI:** 10.3390/pharmaceutics14102057

**Published:** 2022-09-27

**Authors:** Vyara Velcheva, Kaspar Hegetschweiler, Georgi Momekov, Stefka Ivanova, Angel Ugrinov, Bernd Morgenstern, Galina Gencheva

**Affiliations:** 1Faculty of Chemistry and Pharmacy, Sofia University “St. Kliment Ohridski”, 1 J. Bourchier Blvd., 1164 Sofia, Bulgaria; 2Fachrichtung Chemie, Universität des Saarlandes, Campus, D-66123 Saarbrücken, Germany; 3Department of Pharmacology, Pharmacotherapy and Toxicology, Faculty of Pharmacy, Medical University of Sofia, 2 Dunav Str., 1000 Sofia, Bulgaria; 4Department of Pharmaceutical Chemistry and Pharmacognosy, Faculty of Pharmacy, Medical University of Pleven, 1 St. Kliment Ohridski Str., 5800 Pleven, Bulgaria; 5Department of Chemistry and Biochemistry, North Dakota State University, 1311 Albrecht Blvd., Fargo, ND 58102, USA

**Keywords:** Pt(IV) complexes, synthesis and characterization, biological activity, cisplatin resistance, structure-activity studies, novel metal-based drugs

## Abstract

Metal complexes occupy a special place in the field of treatment and diagnostics. Their main advantages stem from the possibility of fine-tuning their thermodynamic properties and kinetic behavior in the biological milieu by applying different approaches such as properly constructed inner coordination sphere, appropriate choice of ligands, metal oxidation state, redox potential, etc., which are specific to these compounds. Here we discuss the design and synthesis of two octahedral cationic Pt(IV) complexes of the tridentate ligand all-*cis*-2,4,6-triaminocyclohexane-1,3,5-triol (taci) with composition, *fac-*[Pt(taci)I_3_]^+^, **1** and *bis-*[Pt(taci)_2_]^4+^, **2** as well as the potential for their application as antineoplastic agents. The complexes have been isolated in a solid state as: *fac-*[Pt(taci)I_3_]I·3H_2_O (**1A**), *fac-*[Pt(taci)I_3_]I (**1B**), *fac-*[Pt(taci)I_3_]I·2DMF (**1C**), *bis-*[Pt(taci)_2_](CO_3_)_2_·6H_2_O (**2A**) by changing the acidity of the reaction systems, the molar ratios of the reagents and the counterions, and by re-crystallization. The ligand taci is coordinated through the NH_2_-groups, each molecule occupying three coordination places in the inner coordination sphere of Pt(IV). Monitoring of the hydrolysis processes of **1A** and **2A** at different acidity showed that while **2A** remained stable over the study period, the I^−^-ions in **1A** were successively substituted, with the main product under physiologically mimetic conditions being *fac,cis-*[Pt(taci)I(OH)_2_]^+^ (**h2**). The antiproliferative tests involved eight cancer cell models, among which chemosensitive (derived from leukemias and solid tumors) and chemoresistant human Acute myeloid leukemia lines (HL-60/Dox, HL-60/CDDP), as well as the non-malignant kidney’ cells HEK-293T showed that the complexes **1A** and **2A** are characterized by a fundamentally different profile of chemosensitivity and spectrum of cytotoxic activity compared to cisplatin. The new Pt(IV) complexes were shown to be more effective in selectively inhibiting the proliferation of human malignant cells compared to cisplatin. Remarkable activity was recorded for **1A**, which showed an effect (IC_50_ = 8.9 ± 2.4) at more than 16-fold lower concentration than cisplatin (IC_50_ = 144.4 ± 9.8) against the resistant cell line HL-60/CDDP. In parallel, **1A** exhibited virtually the same cytotoxic effect against the parental HL-60 cells (IC_50_ = 9.0 ± 1.2), where cisplatin displays comparable chemosensitivity (IC_50_ = 8.3 ± 0.8). The determined resistance indices (RI~1) show unequivocally that the resistant lines are sensitive to both compounds tested; therefore, they are capable of overcoming the mechanisms of cisplatin resistance. The structural features of these compounds and their promising pharmacological properties justify their inclusion in the group of “non-classical metal-based antitumor compounds” and are a prerequisite for the admission of alternative mechanisms of action.

## 1. Introduction

The idea of using metal complexes as medicines [1] and, in particular, for the treatment of cancer has been known for a long time. However, a milestone in this field of research has been the discovery of the effective antitumor properties of the simple inorganic platinum(II) coordination compound, cisplatin (*cis*-diamminedichloro-platinum(II), Cis-DDP) [2] (Figure 1a) and its validation by subsequently studies as an effective anticancer agent in humans [3]. Today, approximately fifty years after the introduction of cisplatin as a drug, platinum-based antitumor coordination compounds (such as carboplatin, oxaliplatin, etc.) are of great importance in the treatment of cancer, and their clinical application increase the prospects for survival of many cancer patients [4]. The emerging drawbacks during the medical application of platinum cytostatics, such as general toxicity and severe side effects [5,6,7], as well as intrinsic and acquired resistance [8,9], justify the efforts to create new coordination compounds with improved properties in terms of expanding the spectrum of chemotherapy and better clinical profile. It is clear that, to meet these requirements, new metal-based drugs need to be designed based on a formula with completely novel and different structural features [10,11].

In the search for new antitumor drugs, special attention has been paid to the higher-valent platinum complexes, such as those of platinum(IV) [12,13,14,15,16] (Figure 1b) and platinum(III) [17,18,19,20]. The antitumor potential of platinum(IV) compounds [21,22,23,24] was recognized along with the discovery of the biological properties of cisplatin [25], but their clinical trial began later [26,27,28]. Regardless of the strategies used to construct these compounds, none of them has yet been approved for worldwide application [10,29]. It is widely accepted that platinum(IV) complexes are promising as prodrugs due to their relative inertness and low toxicity outside and higher activity within the tumor cells as a consequence of reduction to the corresponding square planar platinum(II) analogs. Most of the clinically tested platinum(IV) complexes are composed of an equatorial core that coincides with known platinum(II) drugs (Figure 1b) and two ligands located axially with respect to the coordination plane [10,21,24,30]. Their mechanism of action is believed to involve activation by a two-electron reduction in tumor cells to the corresponding cytotoxic-active Pt(II) drugs and loss of the axial ligands. These ligands can be further used to adjust properties, such as lipophilicity and reduction potential, and also allow the attachment of additional functional or targeting groups [10,29,30,31]. It has recently been shown that if these ligands have their own biological properties, they can affect synergistically or at least additively the action of the platinum(II) drug [32,33,34,35]. In some cases, the axial ligands are selected with properties that provide effective targeting to enzymes, proteins, and hormones involved in the mechanisms of carcinogenesis, thus attacking the specific characteristics of cancer to alter resistance pathways [32,36,37,38].

Ligands that define the coordination plane of the Pt(IV) complexes are also able to alter and improve the spectrum of antitumor activity [10,39,40,41]. The *cis*-located typically N-containing donor ligands, known as “non-leaving” group ligands (or carrier ligands), determine the thermodynamic stability of platinum species, guide ligands substitution reactions, affect the reduction potential [39,40] and the kinetic of hydrolysis of the antitumor agent [41], and they also remain in the formed platinum DNA-adducts [42]. The nature of N-containing ligands is an important factor for both the Pt(II) and Pt(IV) complexes. For example, the DACH ligand (*trans*-R,R-diaminocyclohexane, (Figure 1a) that substitutes the NH_3_-ligands provokes several significant conformation differences [43], and despite the similarity between cisplatin- and oxaliplatin-GpG adducts (1,2-intrastrand cross-link between two adjacent guanosine residues), the two antitumor drugs display different cytotoxicity. [42,43,44] The isomeric form of the DACH ligand, *cis*-1,4-DACH incorporated as a -bidentate N-containing ligand in the promising antitumor drug, kiteplatin (Figure 1a) substantially alters the cisplatin spectrum of activity [45] and thus a real circumventing of the cisplatin cellular self-defense mechanism has been achieved. Instigated by this, a Pt(IV) derivative of kiteplatin has been synthesized to obtain a prodrug with selective tropism towards bone cancer [46]. The most successful Pt(IV) coordination compound, satraplatin (Figure 1b), which meets most requirements, such as sufficient stability to provide oral administration, activation in cancer cells, etc., has been shown to overcome cisplatin resistance. It was found that its active [Pt^II^(ammine)(Cyclohexylamine)]^2+^ reduced species can form two isomeric 1,2-d(Gp) intra-strand cross-links [47]. The ability of satraplatin to overcome cisplatin resistance is thought to be due to the asymmetric nature of DNA lesions [10,47], which is a result of the different nature of the two “carrier” N-ligands. The two last ligands of the Pt(IV) coordination octahedron, located in a *cis*-position, usually anionic monodentate or chelating ligands (so-called “leaving” group ligands), can also be used to correct certain physicochemical properties. These ligands hydrolyze [41,48] and are substituted in the formation of DNA adducts. Modifying these ligands can alter the rate of reduction of the Pt(IV) compounds, aquation kinetics, and, ultimately, drug reactivity. Complexes are considered to be more toxic if these ligands are more labile, which favors their indiscriminate substitution by non-target biological nucleophiles.

Another approach to the construction of antitumor agents based on Pt(IV) complexes is to use ligands that request photoactivation in the biological environment [49], and in most cases, these ligands occupy the positions of the “leaving” ligands. The development of such compounds has many advantages, and most importantly, the formation of cytotoxic species occurs only after irradiation, when the compound is delivered selectively to cancer cells. Among the few examples of photoactivated platinum(IV) prodrugs, the first tested complexes were based on iodide ligands [50]. Despite the few positive results, studies on *cis*-diiodo Pt(IV) complexes as photoactivated agents have been suspended due to their common toxicity in the dark and reduction reactions by biological thiols. However, recently the focus has been again on platinum-iodide complexes as active anticancer agents [51], and they have become pharmacologically prospective due to the effect of the iodide ligands that are responsible for a significant shift in various mechanistic aspects, including hydrolysis, reactivity towards relevant biomolecules, cell cycle modification or induction of apoptosis [52].

The design of anticancer drugs is usually guided by knowledge of the mechanisms of their biotransformation and interactions in the biological environment [53]. Besides the covalent binding of metal-based drugs to biomolecules, the mechanism based on “activation by reduction” has successfully worked for Pt(IV) complexes. In addition, a series of papers addressed another redox mechanism based on direct oxidative damage of DNA provoked by a Pt(IV) compound. Thus, tetraplatin (Figure 1b, [Pt^IV^(DACH)Cl_4_]) converts guanine to 8-oxo-guanine with simultaneous reduction to [Pt^II^(dach)Cl_2_] [54]. Studies on the mechanism and kinetic of oxidation of guanine bases in double-stranded oligonucleotides showed that the Pt(IV) complex binds to G-N(7), and the bonding is followed by a two-electron inner sphere transfer from the guanine bases to Pt(IV) [55,56,57].

The accumulated data on the influence of the structural characteristics of platinum cytostatics on their physicochemical and biological properties, as well as knowledge of possible mechanisms [10] of their cytotoxic behavior, are indisputably useful to guide in the early stages of metal-based drug development. All factors are important here, such as the oxidation state of platinum, the nature, acid-base properties, and denticity of the ligands from each group, the spatial arrangement of the donor atoms around the metal, etc. There are excellent examples, such as pyriplatin [58] and phenanthriplatin [59], which demonstrate the potential to avoid common mechanisms of cellular resistance [60]. This is believed to be due to their structure and, in particular, their inner coordination sphere, which completely violates the classical activity rules.

In the present study, we propose a novel drug design approach based on octahedral Pt(IV) coordination with a carrier ligand that occupies three positions simultaneously via its three amino groups, equivalent in character. Applying it consistently, two octahedral cationic Pt(IV) complexes of the tridentate ligand all-*cis*-2,4,6-triaminocyclohexane-1,3,5-triol (taci) were obtained, namely with one coordinated taci (taci:Pt = 1, taciplatin) and with two coordinated taci molecules (*bis-*compound, taci:Pt = 2, ditaciplatin), Figure 1c. A peculiarity of taciplatin is that its inner coordination sphere is supplemented by three monodentate I^-^ ions as “leaving” ligands. The carrier ligand, known by the trivial name taci [61], can be considered a 1,3,5-triamino-analogue of *cis*-inositol that is a stereoisomer of the biomolecule *myo*-inositol [62]. The latter compound and its derivatives have been discussed in medicinal research in the treatment of ailments from diabetes to cancer [63,64]. Although the pharmacology of inositols is relatively unknown, advantages could be sought if its 1,3,5-triamino analog is used to design antitumor drugs. From the point of view of synthetic chemistry, taci demonstrates a remarkable molecular structure defined by a rigid cyclohexane ring with two types by nature, alternating successively substituents [61], and the behavior of a tridentate ligand. Two of its chair conformers, namely these with the *syn*-1,3,5-triaxial arrangement of OH- or NH_2_-groups, are the most stable. In these, the ligand provides four different metal ions binding modes, all of which are performed as facial coordination. Metal ions characterized by different charges and sizes choose an appropriate manner to bind to the ligand according to the specific steric requirements and electronic properties of the individual coordination site [61]. Analyzing most of the published data on the coordination properties of the ligand [65,66,67,68,69,70,71,72], it was found that due to its flexible nature, it forms stable complexes with almost all metal ions from Li(I) to Bi(III), and only the data about the platinum metals are scarce. In this paper, we discuss the synthesis, structure, and in vitro antineoplastic activity of the two novel platinum(IV) complexes of the ligand taci and evaluate the effect of the inner coordination sphere on their biological properties.

## 2. Materials and Methods

### 2.1. Chemicals and Physical Measurements

Initial reagents for synthesizing the complexes: K_2_PtI_6_ was prepared by a method described in the literature [73]. The ligand taci hydrate (taci·2H_2_O), freshly prepared from its trihydrochloride or sulfate precursor by ion exchange chromatography (Dowex 1, OH-form) [74], was used for all synthetic procedures. All other chemicals and reagents were purchased from Sigma-Aldrich Chemie GmbH, Steinheim am Albuch, Germany; Merck, Darmstadt, Germany and Edelmetall–Chemie, Freiberg, Germany and were used without further purification. All solvents used were of analytical grade.

NMR spectra were obtained using Bruker Avance III HD spectrometer (500.13 MHz for ^1^H, 125.79 MHz for ^13^C NMR, 50.67 MHz for ^15^N, and 107.51 MHz for ^195^Pt). The measurements were made of samples dissolved in D_2_O using 3-(Trimethyl-silyl) propionic-2,2,3,3-d_4_ acid sodium salt as the internal standard for ^1^H and ^13^C spectra (δ = 0 ppm) and K_2_PtCl_4_ as the external standard for ^195^Pt (δ = −1612.81 ppm). Water suppression was performed by the standard Bruker pulse program ***noesygppr1d***, which uses supersaturation during relaxation delay and mixing time and spoil gradients. For the ^13^C and ^15^N CP-MAS (cross-polarization magic angle spinning), NMR spectra, commercially available solid-state double resonance probes supporting zirconia rotors with 2.5 mm outer diameter were used, and Standard CP-MAS pulse sequences were applied. ^13^C and ^15^N spectra were referenced to glycine (δ(^13^C) 43.58 and δ(^15^N) 33.37 ppm) as an external standard. For ^1^H and ^195^Pt 1-pulse pulse sequences were used with K_2_PtCl_6_ as the external standard. IR spectra were recorded on a Nicolet 6700 FTIR Spectrometer from ThermoFisher Scientific in the 4000–400 cm^−1^ spectral region (32 scans, 2 cm^−1^ resolution) and an INVENIO-R Bruker FTIR spectrometer in the 6000–80 cm^−1^ spectral region (100 scans, 2 cm^−1^ resolution) in KBr and CsI pellets. The High Resolution-Electrospray Ionization Mass Spectrometry (HR-ESI MS) analyses were performed on the Waters SYNAPT G2-Si Q-ToF system. All samples were dissolved in water (Optima, Fisher Chemical, MS-grade) at the appropriate acidity of the environment and injected directly (no LC columns and methods used). All observations were made with resolution mode (20 kDa resolution) and m/z range 50 to 2000 Da with the following ESI settings: capillary voltage 3.0 kV, samples cone voltage 40 V, source temperature 90 °C, desolvations temperature 250 °C, desolvation gas flow 350 L/h. LockMass correction with leucine encephalin was applied for all samples for better mass accuracy. Elemental analysis was performed on EuroEA3000 CHNS-O Analyzer. The pH measurements were made using a Metler Toledo “Seven Compact” pH meter equipped with a combined InLab Micro Pro pH-electrode.

### 2.2. Synthesis and Characterization

*fac-*[Pt(taci)I_3_]I·3H_2_O, **1A**. Aqueous solutions of K_2_PtI_6_ (0.4569 g, 0.44 mmol dissolved in 15 mL) and taci·2H_2_O (0.068 g, 0.29 mmol dissolved in 10 mL) were mixed at a molar ratio Pt:taci = 1.5:1. The mixture was stirred for 3 h while heating at 60 °C. The obtained red crystalline precipitate was filtrated and recrystallized (pH~5). Red crystals of **1A** crystallized after a few hours. Elemental analysis, calcd (%) for I_4_C_6_H_21_N_3_O_6_Pt^IV^, **1A**, (933.91 g·mol^−1^): C 7.72, H 2.26, N 4.50; found: C 7.61, H 2.22, N 4.44. IR (KBr disks) [cm^−1^]: 3443, 3358, 3248, 3225, 3171, 2923, 2853, 1621, 1549, 1529, 1514, 1467, 1359, 1332, 1304, 1282, 1208, 1179, 1151, 1072, 1046, 938, 867, 799, 726, 700, 625, 572, 518, 481, 435, 415. **^1^H-NMR**, δ [ppm] (*J* [Hz]): 3.29^(a)^ (3H, t, ^3^*J*_HaxHeq_: 4.42, ***H***-(C-NH_2_)); 3.29^(b)^ (3H, dt, ^3^*J*_PtH_: 47.0, ^3^*J*_HaxHeq_: 4.42, ***H***-(C-NH_2_)); I(3.29^(a)^:I(3.29^(b)^ = 2 (determined by ^195^Pt—33.8%); 4.43 (3H, t, ^3^*J*_HeqHax_: 4.43, ***H***-(C-OH)); **^13^C{1H}-NMR**, δ [ppm] (*J*[Hz]): 48.92 (^2^*J*_PtC_: 19.07, ***C***-(NH_2_-Pt)); 61.41 (^3^*J*_PtC_: 14.98, ***C***-(OH-C(NH_2_-Pt)); **HR-MS [ESI]:** [**1**^+^] 752.7895 (calcd 752.7890).

Crystals suitable for single-crystal X-ray diffraction analysis of *fac-*[Pt(taci)I_3_]I (**1B**) and *fac-*[Pt(taci)I_3_]I·2DMF (**1C**) were obtained as follows: for **1B**: 0.001 g of **1A** were covered with liquid paraffin oil (Nujol) and several days after crystals of **1B** were obtained; for **1C**: 0.001 g of **1A** were dissolved in 2 mL DMF (N,N-dimethylformamide) and red crystals of **1C** were grown by slow diffusion of ether into the solution.

*bis-*[Pt(taci)_2_](CO_3_)_2_·6H_2_O, **2A**. The solution of K_2_PtI_6_ obtained by dissolving 0.1699 g (0.16 mmol) in 5 mL distilled water at stirring was mixed with a 5 mL aqueous solution of taci·2H_2_O (0.0739 g, 0.32 mmol, pH = 8) at a molar ratio taci:Pt=2 and the mixture was stirred continuously for an hour at slight heating (t < 40 °C) until the formation of a brown precipitate. The heterogeneous mixture was then heated at 60–70 °C until the brown precipitate became lighter and the pH of the solution reached ~8.5. Then a solution of 0.1 M NaOH was added to pH~9.0–9.5, and the mixture was heated at 60–70 °C for 1-2 h. Yellow crystals suitable for single crystal X-ray diffraction analysis were obtained 1 day after. Elemental analysis, calcd (%) for C_14_H_42_N_6_O_18_Pt, **2A**, (777.51 g.mol^−1^): C 21.63, H 5.434, N 10.81; found: C 21.77, H 5.055, N 11.24. IR (KBr disks) [cm^−1^]: 3438, 3155, 3073, 3015, 2926, 2844, 2705, 2657, 1606, 1572, 1473, 1410, 1394, 1372, 1347, 1240, 1172, 1073, 976, 966, 887, 821, 599, 539, 473. **^1^H-NMR**, δ [ppm] (*J*[Hz]): 3.04^(a)^ (3H, t, ^3^*J*_HaxHeq_: 4.30, ***H***-(C-NH_2_)); 3.04^(b)^ (3H, dt, ^3^*J*_PtH_: 42.3, ^3^*J*_HaxHeq_: 4.30, ***H***-(C-NH_2_)); I(3.04^(a)^:I(3.04^(b)^ = 2 (^195^Pt—33,8%); 4.33 (3H, t, ^3^*J*_HeqHax_: 4.30, ***H***-(C-OH)); **^13^C{1H}-NMR**, (δ [ppm] (*J*[Hz]): 50.50 (^2^J_PtC_: 18.17, ***C***-(NH_2_-Pt)); 63.47 (^3^*J*_PtC_: 11.81, ***C***-(OH-Pt)). **HR**-**MS [ESI]:** [(**2**-3H)^+^] 546.1643 (calcd 546.1641).

### 2.3. Studies on the Stability of the Complexes in Solution

The investigations on the stability of the complexes in solution were performed by means of NMR spectroscopy and mass spectrometry. The ^1^H NMR experiments were conducted as samples of **1A** (1.1 × 10^−3^ mmol) and **2A** (1.3 × 10^−3^ mmol) were dissolved in 1 mL D_2_O solutions. Mass spectra were recorded in water solutions (1 mg/L) at room temperature. The proper pH values were achieved by adding 0.01 M NaOH (NaOD for NMR). The experiments were also conducted in commercially available PBS (Phosphate-buffered saline, Merck, pH = 7.4). The reactions were followed immediately after sample dissolving and every 24 h in a 72 h period. In order to closely simulate the human body’s cellular conditions, the solution behavior of **1A** and **2A** in PBS (pH = 7.4) were also monitored at 37 °C by ^1^H-NMR.

### 2.4. Single-Crystal X-Ray Diffraction

The best appropriate single crystal from each of the analyzed samples was selected and mounted on a micro loop under a microscope for further data collection. The intensity data were collected on a Bruker SMART X2S benchtop system with an air-cooled micro-focus Mo X-ray source (Kα λ = 0.71073) and a BREEZE CCD detector at room temperature. The data were processed, including numerical absorption correction, with Apex 2 package [75,76]. The structures were solved by Intrinsic Phasing (SHELXT) [77], and non-hydrogen atoms were refined by a full-matrix least-squares procedure using anisotropic displacement parameters (SHELXL) [78] with graphical user interface software OLEX 2 [79]. All hydrogen atoms in the studied structures were placed in calculated positions and treated using a riding model with fixed isotropic thermal displacement parameters (1.2 or 1.5 times those of the respective atom). **1A** exhibits substitutional disorders over two water molecules (O4S and O5S), which were refined with occupancy of 0.33 and 0.50, respectively. The delocalized solvent’s molecules in **1A** were handled with the SQUEEZE function of PLATON software [80], removing an electron density of 80 electrons per a crystallographic cell. Two disorders are resolved in the **2A** refinement. The C7 atom of one of the carbonate anions lies on a 2-fold rotation axis, but the attached O4 atom is slightly off of it; thus, its position was solved by a disorder about a special position around this axis (0.50 occupancy). The carbon atom C8 from the second carbonate lies at an inversion center, and each of its three oxygen atoms has an image, thus occupies 2 opposite positions with 0.50 occupancy. The Olex 2 [79] and Mercury [81] software were used to prepare publication materials, such as CIF files and pictures. Crystals’ descriptions, data collection specifics, and refinement statistics for all compounds are given in Appendix A. All crystallographic data were deposited at the Cambridge Crystallographic Data Centre with numbers: 2059934 (**1B**), 2059935 (**1C**), 2059937 (**2A**), and 2094053 (**1A**).

### 2.5. Cell Lines, Culture Conditions, and Cytotoxicity Assay

The human tumor cell lines used in this study, namely HL-60 (acute myeloid leukemia), SKW-3 (T-cell leukemia), LAMA-84 (chronic myeloid leukemia in blast crisis), SAOS-2 (osteogenic sarcoma), MDA-MB-231 (breast adenocarcinoma) and T-24 (urinary bladder carcinoma) were purchased from German Collection of Microorganisms and Cell Cultures (DSMZ GmbH, Braunschweig, Germany). The potency in overcoming the resistance effects from the new compounds was tested on HL-60/Dox (doxorubicin-resistant) [82] and HL-60/CDDP (cisplatin-resistant) [83] sublines. The multidrug-resistant derivative line HL-60/Dox was obtained from the German Cancer Research Center DKFZ in Heidelberg, Germany. The induced multidrug-resistant phenotype has been sustained through cultivation in a growth medium of 0.2 μM doxorubicin. The growth medium was maintained free of the anticancer agents at least 5 days before the experiments. The HL-60/CDDP subline was developed at the Laboratory of Experimental Chemotherapy (Faculty of Pharmacy, MU-Sofia). The selection was made through long-term serial exposures of the HL-60 stem cell line to gradually increasing concentrations of cisplatin. The cisplatin-resistant phenotype has been sustained through cell cultivation in a growth medium containing 25 μM cisplatin. Resistant cells were incubated in a platinum-free environment at least 5 days prior to the experiment to avoid possible synergistic interactions with the compounds being screened for cytotoxicity. In vitro nephrotoxicity testing was performed on a human embryonic kidney cell line HEK-293T. The cells were maintained in a controlled environment—cell culture flasks at 37 °C in an incubator ‘BB 16-Function Line’ Heraeus (Kendro, Hanau, Germany) with humidified atmosphere and 5% CO_2_. They were kept in the log phase by supplementation with fresh medium, 2 or 3 times a week. The cells were grown in RPMI-1640 medium supplemented with 10% fetal bovine serum and 2 mM L-glutamine. The tumor cell growth inhibitory effects were assessed using the standard 3-[4,5-dimethylthiazol-2-yl]-2,5-diphenyl-2H-tetrazolium bromide (MTT)-dye reduction assay as described by Mosmann with minor modifications [84,85]. Exponentially growing cells were seeded in 96-well flat-bottomed microplates (100 μL/well) at a density of 1×10^5^ cells per mL, and after 24 h incubation at 37 °C, they were exposed to various concentrations of the tested compounds for 72 h. For each concentration, at least 8 wells were used. After the incubation with the test compounds, 10 μL MTT solution (10 mg/mL in PBS) aliquots per well were added. The microplates were further incubated for 4 h at 37 °C, and the MTT-formazan crystals formed were dissolved through the addition of 100 μL/well 5% formic acid (in 2-propanol). The absorption was measured using a microprocessor-controlled microplate reader at 580 nm. The cell survival data were normalized to the percentage of the untreated control and were fitted to sigmoidal dose/response curves, and the corresponding half of inhibition concentration values were calculated using non-linear regression analysis. The cell growth inhibition was determined by triplicate assays. The half of inhibition concentrations (IC_50_) values were calculated from the cytotoxicity curve. The activity of the tested new Pt(IV) compounds to the produced cisplatin-resistant sublines HL-60/Dox and HL-60/CDDP was evaluated by the resistance index (RI) and compared with the RI of the parent drug cisplatin. The RI for each platinum compound (**1A**, **2A**, and cisplatin) was calculated via dividing the IC_50_ determined for the cisplatin-resistant sublines HL-60/Dox and HL-60/CDDP by the IC_50_ determined for the parent HL-60 cells. A comparative evaluation of the activity of each of the novel compounds and cisplatin for a given cell line in terms of its effect on all other cell lines was performed using the method developed by National Cancer Institute, Bethesda, MD, USA. In brief: The IC_50_ values characterizing the effect of the particular drug, determined for all cell lines selected in the panel, were averaged, and the mean value obtained was presented as a logarithm (log(av.IC_50_)). The log(IC_50_) value determined of this compound in the individual lines is compared with the logarithm of the mean value, and the results obtained are summarized on a “mean graph” diagram. The selectivity of the tested compounds towards the malignant cells included in this study was analyzed by correlating the IC_50_ values determined in the HEK-293T and these determined in MTT experiments for the malignant cells. The results were presented as an index of selectivity (IS) which was calculated as a ratio of the IC_50_ value determined by the MTT experiment in the HEK- 293T cell line and the arithmetic mean of the IC_50_ values determined in the malignant cells for each of the studied complexes including cisplatin.

## 3. Results and Discussion

### 3.1. Synthesis

An aqueous solution of K_2_PtI_6_ was used as an initial Pt(IV) compound for the synthetic procedures. The hydrolysis of the hexaiodoplatinate(IV) takes place in three stages [86] though successive substitution of three of all six iodide ions from the inner coordination sphere of platinum(IV) with water molecules. Further substitution of the remaining iodides is hampered by the ongoing parallel redox processes. It is thought that in the presence of taci·2H_2_O introduced as a fresh aqueous solution into the system, and the complexation reaction occurs with the substitution of the water molecules of the hydrolysis product [Pt^IV^I_3_(H_2_O)_3_]^+^ of the platinum salt with donor atoms from taci (Figure 2). The interaction starts in a slightly acidic medium (pH value is in the interval 4.5–6.0) when heated (60 °C). The brown precipitate formed at the beginning upon the interaction of the reagents in equimolar quantities gradually lightens, and the dark yellow solution becomes clear. Red crystals were obtained after several hours from this solution at room temperature. The elemental analysis data are consistent with a composition of 1:1 Pt-taci for the complex, [PtI_3_(taci)]I·3H_2_O, **1A**. The great stability and low reactivity of Pt-I bonds is the reason for the coordination of only one molecule of the tridentate ligand taci through substitution, most probably of the three water molecules from the inner coordination sphere of Pt(IV). The charge of the obtained cationic complex [PtI_3_(taci)]^+^, 1 is neutralized by an iodide liberated during the hydrolysis of the starting platinum salt.

Further on, the 1:2 Pt-taci complex, 2, was obtained after the addition of 0.1 M NaOH to the dark yellow solution of the reaction system to achieve a pH value of 9.0–9.5 under heating. During this second stage of the interaction, the solution turned light-yellow, and a few hours later, yellow crystals were obtained with a composition [Pt(taci)_2_](CO_3_)_2_·6H_2_O, **2A**. It appears that in order to coordinate the second taci molecule, it is necessary to substitute the three iodides from the inner coordination sphere of complex 1 with hydroxido ligands (Figure 3) present in excess in the second stage of the interaction due to increased pH through addition of NaOH. Now the coordination of the second taci molecule by substituting the hydroxido ligands is favored by the *trans*-influence of the Pt-N bonds from the inner coordination sphere of complex 1, leading to the weakening of the Pt-OH bonds.

A peculiarity in obtaining the complex **2** in a solid state as **2A** is the presence of CO_3_^2-^ ions in the reaction system. As an alicyclic primary triamine, in an aqueous medium at pH > 6.0, the ligand taci interacts with CO_2_ [87] from the air. The equilibria that are established with increasing pH give the successive products: carbamic acids, carbamates (as can be seen from the ^13^C-NMR spectrum of the free ligand, recorded at pH = 8.4, Appendix A), bicarbonate ions and at pH > 9.5 upon heating, the carbonate ions required to isolate complex **2** in a solid state as **2A** are released. Studying in detail the reaction of PtI_6_^2-^ and taci under different reaction conditions, it was found that the leading factor that directs the syntheses is the acidity of the reaction system. The proper pH values can be achieved based on the acid-base properties of taci [74] by adjusting the molar ratio Pt:taci. Thus, **1A** was directly obtained at a molar ratio Pt:taci = 1.5 and the complex **2A** at molar ratio taci:Pt = 2 with an additional calculated quantity of NaOH. Complex **1** was also isolated in a solid state as *fac-*[PtI_3_(taci)]I, **1B**, where the water molecules have left the crystal lattice of **1A**. The compound **1B** was obtained by crystal-to-crystal transformation from **1A** under liquid paraffin oil. Complex **1** was also isolated as crystals with two molecules of DMF as *fac-*[PtI_3_(taci)]I·2DMF, **1C** after re-crystallization of **1A** in DMF.

### 3.2. Solid State Characterization

Projections of the complex cations: *fac-*[PtI_3_(taci)]^+^, **1** and *bis-*Pt(taci)_2_]^4+^, **2** with the atom numbering are shown in Figure 1. The crystal packing within the unit cell for each of the studied compounds is shown in Appendix A. Appendix A lists the crystallographic data and structure refinement parameters. The relevant bond lengths and bond angles are represented in Appendix A.

**1A** crystallizes in a monoclinic crystal system, space group P2_1_/c (Appendix A). The asymmetric unit of the compound comprises two complex cations: *fac-*[PtI_3_(taci)]^+^ (noted as **1A**-**A** and **1A**-**B**)), two I^-^ anions, and a set of co-crystallized water molecules. The ligand taci is coordinated as a neutral molecule, and the formation of three identical six-membered chelate rings as Pt-N-C-C-C-N (average of the bond angles: Pt-N-C-116.3°, N-C-C-110.7°, C-C-C-113.6°) in the complex cation (Figure 1) is favored by its chair conformation with three axial NH_2_-groups. The mono-charged complex cation adopts six-coordinated geometry with 3 I^−^ and three amino N-atoms of the tridentate taci ligand in the inner coordination sphere. Each amine group of taci is located opposite one iodide ligand, and thus an octahedral geometry with facial PtI_3_N_3_ coordination is formed. The platinum atom is at the center of the octahedral environment. The mean Pt-I (2.640 Å) and Pt-N(amine) (2.107 Å) bonds (Appendix A) for the two complex cations in the crystallographic unit are comparable with these in a *fac*-derivative of the terdentate diethylenetriamine ligand with three iodine atoms around Pt(IV) [88]. In the complex cations, two of the Pt-N distances are close (Pt-N1, Pt-N2) and slightly longer than the third Pt-N3, and two of the Pt-I distances are close (Pt-I1, Pt-I2) and slightly shorter than the third Pt-I3 distance. Thus, the sum of the bond lengths of the opposite bonds N3-Pt-I3 is less (for A-cation of 0.008 Å and B-cation—0.030 Å) than the other two opposite bonds. However, the tetragonal deformation of the regular octahedral geometry is marginal, which is shown by the ratio of the length of each chosen from the N-Pt-I bonds along the axial axis and the mean values of the N-Pt-I bonds in the coordination plane of the octahedron, that is in the range of 0.994–0.998 Å. The values of the bond angles N-Pt-N ≤ 90° and I-Pt-I ≥ 90° together with those N1-Pt-I2 > 90° and N2-Pt-I1 < 90° for **A**-cation and the inverse ratio for **B**-cation suggests trigonal or rhombic deformation. Since the deviation of the bond angles values is negligible (up to 1.5°) and taking into account the d^6^-configuration of platinum, it is accepted that the observed slight distortion of the regular octahedron is due to the taci geometry and the size and electron-acceptor properties of the coordinated iodide ligands, as well as the presence of water molecules, engaged in H-bonding. Further, because of the slight deviation from the octahedral geometry, a local C_s_ molecular symmetry with a mirror plane through I3PtN3C5(H)C2(H)O1 is expected for the complex cations. The fourth iodide anion is located in the cavity, formed by complex cations and water molecules. The structure is stabilized by an extended three-dimensional network of intramolecular and intermolecular hydrogen bonds, which were exactly determined only for the water molecules with occupancy of 100% (Figure 2a, Appendix A). The compound **1B** crystallizes in triclinic P-1 space group (Appendix A) and consists a complex cation: *fac-*[PtI_3_(taci)]^+^ and an I^−^ anion. In **1B**, the difference between bond lengths Pt-N (Δ 0.0067 Å) and Pt-I (Δ 0.0043 Å) is smaller than this in **1A**, but the deviation of the bond angles from 90° is up to 5° (Appendix A). Hence the observed deformation can be considered as a trigonal distortion imposed by the correspondence between the size of the coordination place offered by taci and the ionic radius of platinum(IV) (0.76 Å). Unlike **1A**, compound **1C** crystallizes with two DMF molecules in a monoclinic crystal system with a P2_1_/n primitive cell (Appendix A). In this compound, the asymmetric unit comprises one complex cation, one I^−^ anion, and two DMF molecules (Appendix A). The crystal structure is additionally stabilized by intermolecular H-bonds formation and also intramolecular H-bonds with solvent participation. The slightly distorted octahedral environment of platinum in **1C** is a consequence again of tetragonal and trigonal distortions.

The complex with composition taci:Pt = 2, **2A** (ditaciplatin) crystalizes in monoclinic C2/c crystal system (Appendix A). In this compound, two neutral taci molecules are coordinated through the NH_2_-groups which occupy the six available coordination sites in the inner coordination sphere of platinum(IV) (Figure 1b), and thus a *bis*-taci complex is formed. Overall, all Pt-N bonds in **2A** are shorter than these of complexes **1A**, **1B**, and **1C**. Two of the Pt-N bonds that each molecule taci provides are almost the same (2.068 Å) and are shorter than the third one (2.078 Å). Since the opposite Pt-N bonds from the two coordinated taci molecules are identical and considering the negligible deviation of the angles up to 1.5°, it is accepted that the complex cation in **2A** adopts a slightly elongated octahedral geometry. The asymmetric unit of **2A** contains half of the complex molecule, half of each of the two counter anions, and three water molecules. In the asymmetric unit platinum center is closely surrounded by two water molecules (O1S and O3S) and one carbonate (C8) by H-bonds: O1S-H1SA-O6, O3S-H3SB-O4, O2-H2-O10, O2-H2-O8, N3-H3A-O8, N3-H3A-O9). The second carbonate (C7) and the third water (O2S) are also engaged with H-bonds: O2S-H2SA-O8 and O2S-H2SA-O9 (Figure 2b). Thus, the structure is stabilized at the expense of a network of intramolecular and intermolecular H-bonds (Appendix A). The 3D packing of the compounds is constructed of alternating planes determined from O1S, C8, Pt1, and O3S, approximately perpendicular (89.89°) to the glide planes where the platinum center is located (Appendix A). The two taci-coordinated ligand molecules are disposed of between these planes.

The correspondence between the single crystal structure and the structure of powder samples of **1A** and **2A** was proved based on their spectroscopic characteristics. The mode of taci coordination was investigated by solid-state FTIR spectroscopy (Figure 3). The data concerning the spectra of the free ligand and complexes **1A** and **2A** have been summarized in Appendix A. The IR spectrum of the uncoordinated taci·2H_2_O (Figure 3) has been interpreted based on its combined function as secondary alcohol and a primary amine, considering that the compound is hydrated. The band assignments to the fundamental modes of vibrations have also been made in accordance with the theoretical optimization calculations using the atoms’ coordinates of X-ray diffraction analysis of taci·2H_2_O [66]. Despite the C_3v_ molecular symmetry of the taci molecule itself, its solid state symmetry is lower, taking into account the participation of the NH_2_- and OH- groups in H-bonds with two water molecules [66]. Therefore, a group of bands was assigned to the main vibrations of each functional group (Appendix A). In general, the number of bands in the spectra of the complexes for the different vibrations corresponds to the lower molecular symmetry in their crystal structure, as well as to the participation of the functional groups in different H-bonds. All bands from the spectra of the complexes, as compared with the free-ligand IR spectrum, were affected by the coordination (Figure 3, Appendix A), but those belonging to the stretching vibrations and deformations of the OH-groups are slightly shifted. For example, the OH-stretching bands are shifted to the higher frequencies because of the different engagement in H-bonding of the coordinated taci molecules. In the spectra of the complexes, two couples of bands were observed for NH_2_-stretching vibrations that are shifted to the lower frequencies. The absorption bands assigned to the scissor deformations, δ(NH_2_), and these to the out-of-plane (wagging) vibrations are shifted to higher frequencies, and the bands assigned to the NH_2_-twisting vibration are shifted to lower frequencies. The observed shifts of the bands originating from the NH_2_-groups in the spectra of the complexes prove unambiguously the symmetrical coordination of the ligand taci through the three NH_2_ groups. The facial coordination of the three NH_2_-groups and the three I^−^ in **1A** was also supported by the sets of three bands for stretching Pt-N vibrations in the region 520–430 cm^−1^ and for Pt-I vibrations in the region 310–250 cm^−1^. The equivalent coordination of six NH_2_ in an octahedral environment from two taci ligands in **2A** was supported by only one observed band for Pt-N stretching at 539 cm^−1^. The presence of planar CO_3_^2−^ (D_3h_) in the outer coordination sphere of **2A** was proved by bands for the four normal modes of vibration (Appendix A).

The measured solid-state NMR spectra were a very useful tool for the structural description of the powder sample of **1A** and **2A**. The resonance signals in the ^15^N CP-MAS spectra of the complexes appeared at lower frequencies (−4.50 ppm for **1A** (Appendix A) and −14.50 ppm for **2A** (Appendix A)) compared with the standard. In each spectrum, one noticeably broadened signal was observed. This is consistent with the coordination of taci through all amino N-atoms and the formation of σ-bonds to platinum(IV) which possesses a high electron density. The downfield shift of the signal from the spectrum of **1A** in respect to this of **2A** is a consequence of the deshielding effect of the coordinated in the inner coordination sphere I^−^-ions because of their electron-acceptor properties. In addition, the ^15^N-signal in the spectrum of **2A** displays a spin-spin coupling with ^195^Pt with a coupling constant ^1^J_PtN_ of 179 Hz (Appendix A). The observed signal confirmed the symmetrical coordination of the two taci molecules by NH_2_-groups. In the ^13^C-CP-MAS spectra of the compounds (Appendix A), two sets of signals were observed assigned to the three-ring C-atoms attached to OH (at 60–64 ppm) and these C-atoms attached to NH_2_- groups (in the region 49–52 ppm). The individual C-atoms have separated signals because of the lower molecular symmetry in the crystal structure. The signals of carbons attached to OH are narrower and well separated. Conversely, the signals of carbons attached to the NH_2_- groups are broadened and thus confirmed again their close location to platinum in the molecules. The presence of two carbonates in **2A** was confirmed from downfield signals at ~168 ppm and 167 ppm. A signal of the coordinated platinum(IV) at −746 ppm was also acquired in the ^195^Pt CP-MAS of **2A**. Overall, the signals in the solid-state spectra of **1A** are broader, and it is assumed that this is a consequence of the effect of coordinated in the inner coordination sphere of three I^-^ ligands together with the three NH_2_ groups.

### 3.3. Stability in Solution

Determining the structure and properties in solution and evaluating the stability and hydrolytic behavior of compounds designed as anticancer agents is an important step in drug development. Here, these studies were done by means of NMR spectroscopy and mass spectrometry. The NMR studies were informative about the molecular symmetry and mode of taci coordination, whereas HR-ESI-MS studies proved the identity of the chemical species in solution. The pH values of the water solutions of **1A** and **2A** (10^−3^ mol/L) are 5.5–6.0 and 9.0–9.5, respectively. The HR-ESI(+)-MS spectra recorded of the fresh water solutions of the compounds (~10^−6^ mol/L) display the specific platinum isotopic pattern. The main signals of the corresponding spectra were detected at m/z: 752.7895 for **1A** and 546.1643 for **2A**, and they are consistent with a 1:1 complex with a composition [Pt(taci)I_3_]^+^, **1** and 1:2 complex—[Pt(taci)_2_-3H]^+^, **2** (Table 1).

The ^1^H and ^13^C-NMR spectra of **1** and **2** were recorded on their fresh D_2_O solutions and were compared with the spectra of the free ligand taci (Figure 4, Appendix A). The ^1^H-and ^13^C-NMR spectra of taci must be interpreted, taking into account its highest possible C_3v_ molecular symmetry in solution [74]. Due to the AA’A’’XX’X’’ spin system, its six ring protons exhibit two triplets in the ^1^H-spectrum (in D_2_O), as one of them belongs to the ring hydrogens attached to C-N and the other—to those attached to C-O. The position of the two signals depends on the acidity of the systems [61]. The pH value of the aqueous solution of taci·2H_2_O is 9.0, and the signals were observed at 2.76 ppm (^3^J = 3.00 Hz) and 3.82 ppm (^3^J = 2.98 Hz). The signals shift to higher frequencies with increasing acidity because of the protonation processes. Similarly, upon coordination to platinum(IV), the two triplets of the coordinated taci molecules also move to higher frequencies (Figure 4) as the data of the chemical shifts, δ [ppm], and coupling constants, ^3^*J* [Hz], are presented in Table 1. According to the higher solution acidity of complex **1**, its two triplets are observed at higher frequencies.

The peculiarity of the spectra of the complexes is the observed doublet of triplets on the main triplet at the lower frequencies (*H*(C-N), with a 1:2 intensity ratio to the main triplet due to the spin-spin coupling with ^195^Pt nuclei. The determined coupling constants ^3^*J*(^195^Pt-^1^H) were 47.0 Hz for 1 and 42.3 Hz for 2 (Table 1), and they prove the close location of C-N-hydrogen (*H*(C-N) atoms to platinum. The signals in the ^13^C-NMR spectra for the C-atoms attached to NH_2_ (*C*-NH_2_) and those attached to OH (*C*-OH) also displayed spin-spin coupling with ^195^Pt with *J*-constants as follows: ^2^*J*(^195^Pt-^13^C) was observed for 1—19.1 Hz and for 2—18.6 Hz and ^3^*J*(^195^Pt-^13^C) at 14.4 Hz and 10.8 Hz, respectively (Appendix A). Hence, in fresh solutions, 1 and 2 adopt C_3v_ and D_3d_ molecular symmetry, respectively, and the taci ligand is coordinated symmetrically in the complexes 1:1 and 1:2 through the three NH_2_-groups.

The hydrolytic processes were followed by ^1^H-NMR spectroscopy in appropriately selected acidity of the medium. While the ^1^H-NMR spectrum of **2A** did not change within 5 days after its dissolving, a series of new signals were detected shortly after the dissolution of **1A** in D_2_O. Four new signals which appeared in the framework of 30 min were observed in addition to the signals of **1** (marked as **1**-**1** and **1**-**2**), Figure 5. Two of them in a 2:1 intensity ratio (**h1**-**1**, **h1**-**1**′) were at lower frequencies and displayed ^1^H-^195^Pt coupling and the other two in a 1:2 intensity ratio (**h1**-**2**′, **h1**-**2**) were in higher frequencies. It was found by integration and confirmed by the ^1^H-^1^H COSY experiment that the four signals belong to one chemical species, and they were assigned to the first hydrolytic species of **1**, namely **h1** (Figure 4).

The substitution of one I^-^ with a solvent molecule (here OH^-^) lowers the molecule symmetry to C_s_, and as a result, two different signals were observed for the ring hydrogens attached to **C**-N and also two different signals for the ring hydrogens attached to **C**-O. It should be emphasized that the signal of *H(*C-N) (**h1**-**1**′) was observed at lower frequencies because the N-atom belongs to NH_2_-group coordinated *trans-* to the OH- ligand compared with this of the hydrogen located in the C-NH_2_-group (**h1**-**1**) that is coordinated *trans-* to I^-^ ligand (Figure 5). A reverse placement from high-field (**h1**-**2**) to low-field (**h1**-**2**′) was observed for the signals of the ring’ hydrogens attached to C-O (Figure 5). The hydrolysis processes in the next two hours continued with the formation of the second hydrolytic species, **h2**, with the substitution of two I^-^ ligands (Figure 4). The lower C_s_ molecular symmetry again gives four signals in its ^1^H-spectrum (Table 1). In the last hydrolytic species obtained in the system, all three iodide ligands are equivalently substituted. Its higher C_3v_ symmetry, like the starting complex **1**, gives two signals in the ^1^H-NMR spectrum, namely for *H*C-N (**h3**-**1**) and *H*C-O (**h3**-**2**). The ^1^H-^1^H COSY experiment proved the correspondence between the individual signals and their belonging to the different complex species. Based on the evaluation of the intensity of the signals, it was found that on the fourth day of the experiment, the main species were: 1—15%, **h1**—50%, **h2**—35%, and the presence of **h3** was very little. A similar experiment performed under physiologically mimetic conditions (PBS and 37 °C) showed the consistent formation of the hydrolytic products and their increasing amount with time. It was found that on the 5^th^ day, the ratio of the different species was as follows: **1**: 5%, **h1**: 7%, **h2**—75%, and **h3**: 13%. Therefore, the main product under these conditions is **h2** and the amount of **h3** increases. The addition of NaOD accelerated the hydrolysis, and in the ^1^H-NMR spectrum in D_2_O immediately after adding NaOD (Figure 5), the composition of the hydrolytic products is as follows: **1**: 25%, **h1**: 30%, **h2**: 35%, and **h3**: 10%.

The identity of the species was proved by HR-ESI-MS spectra, where all of the observed signals with precisely determined m/z values were assigned to the corresponding chemical species (Figure 6, Table 1).

### 3.4. Biological Studies

The cell growth inhibitory effects of the novel platinum(IV) complexes taciplatin (**1A**) and ditaciplatin (**2A**) were evaluated in a set of human malignant cell lines with different chemosensitivity (Table 2). The cell lines included in the experiment originate from leukemias and solid tumors. Two chemoresistant cell lines were also included, HL-60/Dox and HL-60/CDDP. During the screening, cisplatin was tested as a referent control. The experimental data from the antiproliferative tests of the studied platinum(IV) compounds and cisplatin were fitted to the sigmoidal concentration-response curves presented in Figure 7 and were evaluated by the calculated IC_50_ values (Table 2). The experiment was performed in the concentration range of 10–1000 µmol/L, where the new complexes **1A** and **2A** exhibited concentration-dependent antiproliferative activity. The observed difference in cytotoxicity of the two new compounds in respect of cisplatin is related to the nature and arrangement of the ligands in the inner coordination sphere of each of them. The comparison between the two tested new compounds definitely showed the more pronounced antiproliferative activity of **1A**. As a consequence of its own structure, **2A** is characterized by slower kinetics of hydrolysis throughout the experiment and, therefore, lower reactivity. The **1A** complex demonstrated antiproliferative activity comparable to clinically used cytostatic cisplatin against breast adenocarcinoma (MDA-MB-231), bladder carcinoma (T-24), and acute myeloid leukemia (HL-60) cell lines. More importantly, it showed more pronounced inhibition of viability and proliferation compared to cisplatin against the osteogenic sarcoma (SAOS-2) cell line and both resistant cell lines, HL-60/Dox and HL-60/CDDP. Although both compounds showed higher antiproliferative activity against the cisplatin-resistant HL-60/CDDP variant, remarkable activity was reported for **1A**, exhibited at more than 16-fold lower concentrations than that of cisplatin.

The analysis of the spectrum of action of the tested compounds, **1A** and **2A**, compared to that of cisplatin by the COMPARE method (National Cancer Institute, Bethesda, MD, USA), and the corresponding “mean graph” diagrams are depicted in Figure 8. The presented patterns show that the new platinum(IV) complexes exhibit a fundamentally different profile of chemosensitivity and a spectrum of cytotoxic activity compared to cisplatin. Of particular importance is the proven higher chemosensitivity in the tumor models characterized by relative or pronounced resistance to the reference cytostatic, with **1A** showing chemosensitivity to both resistant models and **2A** to the cisplatin-resistant model.

The calculated values of the resistance indices (Table 3) of the studied Pt(IV) complexes determined for the cisplatin-resistant subline HL-60/Dox and HL-60/CDDP tend towards 1. Furthermore, these values are smaller than those determined for cisplatin. Hence, an extremely important aspect of the oncopharmacological studies is that the phenotype of multiple drug resistance in HL-60/Dox affects the newly synthesized complexes to a lesser extent than cisplatin (Table 3). These differences are even more pronounced in the model with induced resistance to platinum anticancer drugs, HL-60/CDDP. The HL-60/CDDP model is characterized by increased levels of reduced glutathione, and higher activity of the GSH-homeostasis enzymes glutathione-S-transferase and glutathione reductase, with a concomitant expression of the MRP-1 efflux pump, which mediates its reduced responsiveness to platinating anticancer drugs [89]. The capability of the novel species to generally overcome this resistance pattern could be ascribed to their more avid reduction due to the enhanced reductive capacity of these cells compared to the parent cell line HL-60. The analysis of the resistance indices in these lines unequivocally shows that the newly synthesized compounds overcome the mechanisms of cisplatin inactivation, which indirectly indicates alternative pharmacodynamic properties.

The in vitro nephrotoxicity was also evaluated by MTT-test in the HEK-293T cell line (Figure 9). In contrast to cisplatin (IC_50_-9.7 μmol/L), the newly synthesized complexes showed less pronounced cytotoxicity to HEK 293T cells, as evidenced by the shift of the constructed “dose-response” curves to higher concentrations and a significant increase in the IC_50_ values as follows 84.6 μmol/L and 127.8 μmol/L for **1A** and **2A**, respectively.

The obtained results were used to evaluate the selectivity of the tested compounds with respect to malignant cells included in this study and were presented as an index of selectivity (IS; Figure 10). For this purpose, the ratio of the IC_50_ value determined by the MTT experiment in the HEK 293T cell line and the arithmetic mean determined of the IC_50_ values in the malignant cells for each of the studied complexes, including cisplatin, was calculated. The IS calculated values were used to assess the selectivity of the studied compounds to the tumor cell lines. The results obtained: 3.8 and 1.4 for **1A** and **2A**, respectively, are more than 14 for **1A** and more than 5 for **2A** times higher than the selectivity index of cisplatin (0.26). Therefore, the Pt(IV) complexes under investigation are much more effective than cisplatin in selectively inhibiting the proliferation of human malignant cells.

The biological studies were performed with the complexes taciplatin (**1A**) and ditaciplatin (**2A**) because of their good water solubility (>1 mg/mL). Both compounds show an undeniably different chemosensitivity profile compared to cisplatin, as quantified by their “mean-graph” charts (Figure 8). The two compounds differ in chemosensitivity and to each other. Their specificity in cytotoxic manifestations is due to the difference in their behavior in aqueous solutions and physiologically mimetic conditions. The *trans*-effect of the coordinated NH_2-_ groups of taci in the complex **1A** provokes the substitution of I^—^ligands and accelerates the hydrolysis processes. As shown by the means of the NMR experiment, the hydrolysis species with substituted iodides **h1**, **h2**, and **h3** were formed sequentially, and as time progressed, their content increased. It has also been established that under physiologically mimetic conditions, the main hydrolysis product is *fac,cis-*[Pt(taci)I(OH)_2_]^+^(**h2**). The architecture of the inner coordination sphere of **h2** can be seen as favorable for a mechanism based on covalent interaction with biomolecules by substituting the two cis-coordinated OH-groups. The unsubstituted I^-^-ligand by means of a *cis*-effect can further favor the processes of substitution, for example, with the nucleic bases, and the side OH-groups of taci can stabilize additionally the adducts formed in the biological environment through H-bonding. It appears that the reduction processes to Pt(II)-active species will not be decisive due to the absence of steric hindrance caused by the bulky axially positioned ligands or the absence of two *trans*-disposed ligands in the inner coordination sphere with high electronegativity. The proposed mechanism seems unacceptable for **2A** since there are no factors provoking fast hydrolysis processes, and its less pronounced cytotoxicity can be explained by a direct redox reaction with biomolecules, here including those that are hallmarks of cancer. It should be emphasized that mechanisms based on direct oxidative damage to DNA or other biomolecules could be possible for both complexes. A distinctive feature of the biological behavior of the two investigated compounds is that each of them shows approximately the same cytotoxic effects on the resistant cell lines included in the study and the cells of the parental lines, as a quantitative measure of these effects is the values of the resistance indexes that were determined close to 1. In addition, both test compounds were shown to be much more effective than cisplatin in selectively inhibiting cancer proliferation, exhibiting toxicity at times higher concentrations against the renal non-malignant cells HEK-293T. Therefore, the new compounds exhibit a fundamentally different mechanism of cytotoxicity, and without a doubt, they possess the potential to overcome the cisplatin cellular self-defense mechanism. These results, together with ongoing preliminary studies that show a cell death pathway distinct from cisplatin, demonstrate the potential of the new Pt(IV) complexes in antitumor drug development.

## 4. Conclusions

In search of a new formulation for antineoplastic drug candidates, two octahedral Pt(IV) complexes of the tridentate ligand all-*cis*-2,4,6-triaminocyclohexane-1,3,5-triol with composition *fac-*[Pt(taci)I_3_]I·3H_2_O and *bis-*[Pt(taci)_2_](CO_3_)_2_·6H_2_O have been synthesized and named as taciplatin and ditaciplatin. The ligand in the complexes is symmetrically coordinated through the NH_2_ groups. The solid-state structure of the compounds was studied using X-ray diffraction, solid-state NMR, and FTIR experiments, and the agreement between single-crystal and powder samples was demonstrated. The new platinum complexes exhibited favorable physicochemical properties in water solutions, and under physiologically relevant conditions, the 1:1 complex, taciplatin, revealed relatively fast hydrolysis with substitution of the inner-sphere I^-^ ligands. Its main hydrolysis product on the fifth day under physiological conditions is *fac,cis-*[Pt(taci)I(OH)_2_]^+^ with two substituted I^-^. In this species, the two *cis*-located OH-ligands occupying an equatorial position provide active centers for further attacks in biological systems. The antiproliferative tests involving eight chemosensitive and chemoresistant cell lines demonstrated the concentration-dependent activity of the new compounds, with the activity of taciplatin being superior. The difference in antineoplastic activity between taciplatin and ditaciplatin appears to be due to the slower hydrolysis kinetics of ditaciplatin, which results from the structure of its inner coordination sphere. More importantly, the two compounds displayed higher activity against the resistant cell lines included in this study, as remarkable activity was reported for taciplatin, which exhibited the effect at more than 16-fold lower concentrations than that of cisplatin against the HL-60/CDDP cell line. The data analysis of the biological studies unambiguously highlights the differences of the presented “applicants” for antineoplastic agents with respect to cisplatin and, in particular, their potential in the treatment of resistant cell lines. Moreover, their specific structural features based on octahedral platinum(IV) complexes of the ligand with favorable properties give reasons for more in-dept biological experiments. In this regard, taciplatin deserves special attention. Its octahedral structure constructed by three N-donors of a tridentate ligand and three I^-^ ligands give advantages in respect to control the kinetic behavior, the strength of the M-L bonds in the biological milieu, as well as for application in photodynamic therapy. The proposed new structure of the inner coordination sphere that the new compounds show, together with their promising pharmacological properties, warrants their inclusion as new proposals in the group of platinum-based non-classical antitumor drugs.

## Data Availability

All data relevant to the publication are included.

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
