# Peer review of "Platinum(IV) Complexes of the 1,3,5-Triamino Analogue of the Biomolecule Cis-Inositol Designed as Innovative Antineoplastic Drug Candidates"

_pharmaceutics, 2022, doi:10.3390/pharmaceutics14102057_

Round 1

Reviewer 1 Report

The manuscript submitted by Velcheva and coworkers shows the characterization, speciation in solution, and the antiproliferative activity of two new Pt(IV) coordination compounds against acute and chronic leukemia, breast, bladder, and sarcoma human tumor cells. They evaluate the cytotoxic effect on HEK-293T cells as a preliminary nephrotoxicity assay (IC50 values of 84 and 127 uM for 1A and 2A, respectively) and show a significant antiproliferative effect on resistant cells HL60/Dox and HL60/CDDP. However, some issues must be addressed before considering the publication of the work. 

Comments:

The abstract should include the tumor cell lines and the IC50 values found. 

The introduction is too long. This section needs s restructuration, focusing on the advantages of Pt(IV) coordination compounds. A synthesis of sections 1.2 and 1.3 could work.

Schemes 2 and 3 could be fused using the chemical structures that illustrate the changes of the Pt(IV) coordination sphere to get the desired products instead the condensed formula. 

The solid-state characterization (X-ray diffraction) is too long and is not used in the next section of the manuscript. Please reduce it to the relevant information. It seems that the most important of the manuscript is the X-ray diffraction structures and not the antiproliferative activity. 

In figure 3 should be identified the bands that the authors mention as indicative of taci coordination. Also, the Pt-N bands should be identified and include the spectra showing the Pt-I bands that the authors mention in lines 630-631. 

I think 13C and 15Nsolid state NMR spectra provide more information about the taci coordination than the IR spectra.

The speciation in the solution shows the hydrolysis processes of 1A. The authors report the transformations occurred within the first hours, on the fourth and fifth days. What about the speciation after 72h (the third day? I consider this information very important due to it is the exposure time during the antiproliferative assays. Particularly with identifying significant changes in the present species between pH 5.5-6.0 and physiological conditions.

Section 3.4 Biological studies lacks of discussion. The authors suggest that differences in the antiproliferative activity found for the tumor cell lines exposed to the compound ¿h1 or h2? (Considering the hydrolisis of 1A) relies in its reactivity. It seem trou for HL60, HL60/Dox, and HL60/CDDP with practically the same IC50 values. But, what can be sais for the other cell lines? The generation of h1 or h2 is the main reason for the IC50 values found on the other cell lines?

Considering the reports of other Pt(II) or Pt(IV) compounds evaluated in these tumor cell lines, which is the proposal for explaining the antiproliferative activities observed in the diverse cell lines evaluated?

Figure 8 should include the values for delta in the x-axis that allow the interpretation of the resistance/sensitivity proposed by the authors. An example can be found here https://doi.org/10.1073/pnas.1517402113

The resistance index discussed in lines 815-824 and table 3 is a substantial result of the work. What can the author say about the similar antiproliferative activity on both resistant cell line HL60/Dox and HL60/CCDP? what can be done [Pt(IV) taci(OH)I2] or [Pt(IV)tac(OH2)I] to produce cell growth inhibition?

 Western Blot images and graphs should be included. The authors identify a slight increase of antiapoptotic protein Bcl-Xl and a decrease of procaspase-8, suggesting a different apoptotic mechanism mediated via the extrinsic cell death pathway. However, caspase 8 is involved in both intrinsic and extrinsic apoptosis pathways. How they could suggest a different apoptotic pathway with this data?

The author must include a proposal of the antiproliferative activity of the evaluated compounds considering the information reported in the literature. Besides, it should be convenient to compare the antiproliferative potency of these compounds with other Pt compounds besides CDDP.

Author Response

Dear Reviewer,

Thank you for your review of our manuscript and for your comments. All of them we have been reviewed and taken into consideration.

Author Response

Dear Reviewer,

Thank you for taking the time to review our work and for your evaluation of the research presented in the manuscript.

All your comments, corrections and suggestions have been reviewed and taken into consideration.

Please  see the attachments for the responses.

Reviewer 3 Report

This paper reports the synthesis and characterization both in solution by NMR, and in the solid state by X-ray diffraction, solid-state NMR and FTIR experiments, of two octahedral cationic Pt (IV) complexes with the all-cis-tridentate ligand 2,4,6-triaminocyclohexane-1,3,5-triol (taci) with composition [Pt(taci)I3]+ and [Pt(taci)2]4+. These complexes were used as anticancer agents against eight chemosensitive and chemoresistant cell lines. The complex with only one tridentate ligand [Pt(taci)I3]+ showed activity higher than [Pt(taci)2]4+. The in vitro pharmacological results were compared with those obtained using cisplatin and they supported a different apoptotic mechanism than cisplatin and its analogues.

The results obtained with these two new platinum complexes are biologically interesting, their synthesis is well described and the characterization of the complexes is complete, so in the opinion of this referee the paper can be accepted for publication in Pharmaceutics.

Author Response

Dear Reviewer,

Thank you very much for taking the time to review our work. We appreciate a lot your review and we are very grateful for your high rating of our studies!

Best regards, Galina Gencheva

Round 2

Reviewer 1 Report

Comment 1. The suggestion is to include the IC50 values of the Pt(IV) compounds in those lines where they show the highest antiproliferative activity.

Comment 2. Literature is plenty of reviews describing the advantages and disadvantages of Pt(IV) coordination compounds as anticancer agents, focusing on antiproliferative activity modifications relying on inner sphere modifications. Therefore, I insist that the introduction should be modified, eliminating the information provided in the different reviews 

https://doi.org/10.1016/j.jinorgbio.2020.111353

https://doi.org/10.3390/ijms22083817

https://doi.org/10.1155/2018/8276139

https://dx.doi.org/10.1021/acs.inorgchem.9b03683

There is no reason to include information that can be consulted in the reviews. 

Comment 3. I agree entirely that the complete characterization of new molecules will be vital in describing and explaining their biological activity, particularly when the structure is retained in both solution and solid state. As mentioned by the author in response to comment 4, the description of the interactions that stabilized compound 1A is only justified in the main manuscript due to the contribution to the description of ligand kinetic exchange described in solution and studied by NMR.   The rest of the crystallographic data analysis of 1B and 2A structures should be moved to supplementary information.  

Comment 7. I agree with the unequivocal identification of product h2 as the main product in the physiological conditions studied in this work. Therefore, the biological assays described in section 4 should be associated with structure h2 instead compound 1. The discussion needs to be modified by indicating that h2 is responsible for the antiproliferative activity.

Comment 8. Considering that h2 was identified as the main product of the hydrolysis of compound 1 in physiological conditions, the authors should provide a proposal about the effect of inner sphere modifications in reactivity and its influence on antiproliferative activity. In my opinion, Pt(IV) inertness could be proposed for [Pt(IV)(taci)(I3)]+ but not for  [Pt(IV)(taci)(I)(OH)2]+; the substitution kinetics of hydroxy ligands must contribute in the biological activity observed. 

Comment 11. I suggest removing data associated with caspase 8 and including it with the results obtained from the ongoing work. Enhance the discussion to explain the results comparing with other compounds from literature.

Comment 12. I agree that much of the information in the introduction could be moved to the discussion section to explain several results obtained. In response to comment 12, the authors mention that they expected that Pt(IV) compuds undergo reduction to Pt(II) species, undergo hydrolysis, and ultimately platinate DNA. However, in table 1 the author proposes hydrolysis of compund 1 to get compound  [Pt(IV)(taci)(I)(OH)2]+ without the redox transformation.

Why do they propose hydrolysis before the redox reaction if they expected the redox process and then the hydrolysis?

Could compound h2 platinate DNA without suffering the redox transformation? 

Does it influence the biological activity of the compound?

What can the authors say about this?

Author Response

Dear Reviewer,

Thank you very much for the Second Edition of your comments on our paper (ID: Pharmaceutics-1842994).

Comment 1. The suggestion is to include the IC50 values of the Pt(IV) compounds in those lines where they show the highest antiproliferative activity.

Response: We could not agree that only one IC50 value for a compound without comparison with this of cisplatin will improve the quality of the abstract. Of course we can include in the abstract the following text in row 34 :

Antiproliferative tests involving eight chemosensitive and chemoresistant cell lines showed the higher activity of the investigated compounds against the resistant cell lines included in this study (HL-60/Dox. HL-60/CDDP), with remarkable activity demonstrated for 1A, which exhibited an effect (IC50=8.9±2.40) at more than 16-fold lower concentrations than cisplatin (IC50=144.4±9.80) against the resistant HL-60/CDDP cell line.

In my opinion, this additional information should not be discussed before analyzing the results.

Comment 2. Literature is plenty of reviews describing the advantages and disadvantages of Pt(IV) coordination compounds as anticancer agents, focusing on antiproliferative activity modifications relying on inner sphere modifications. Therefore, I insist that the introduction should be modified, eliminating the information provided in the different reviews.

https://doi.org/10.1016/j.jinorgbio.2020.111353

https://doi.org/10.3390/ijms22083817

https://doi.org/10.1155/2018/8276139

https://dx.doi.org/10.1021/acs.inorgchem.9b03683

There is no reason to include information that can be consulted in the reviews. 

Response: Thank you for the suggestion, but I couldn't agree with that. The three review examples you noted are primarily focused on the nature of axial ligands alone and specifically on the use of biologically active ligands, and only one of them is included in the introduction and cited as #37. The last example is a perfect experimental work devoted again to the modeling of the axial ligand and here the effect of the oxidation state of platinum is decisive (see the analysis of your examples below).

  • https://doi.org/10.1016/j.jinorgbio.2020.111353

Gibson, D. Platinum(IV) anticancer agents; are we en route to the holy grail or to a dead end? J. Inorg. Biochem. 2021, 217, 111353-111363. https://doi.org/10.1016/j.jinorgbio.2020.111353.

This article is an excellent review devoted to the nature of the axial ligands that determine the chemical and pharmacological properties of Pt(IV) prodrugs. Here, the emphasis is on the approaches used to increase the selectivity of prodrugs to cancer cells and the use of multi-acting prodrugs (combined with bioactive ligands) to achieve a multi-treatment effect and to overcome resistance. and this work is cited in our introduction under No 37.

  • https://doi.org/10.3390/ijms22083817

Daniil Spector, Olga Krasnovskaya, Kirill Pavlov, Alexander Erofeev, Peter Gorelkin, Elena Beloglazkina Alexander Majouga: Pt(IV) Prodrugs with NSAIDs as Axial Ligands. Int. J. Mol. Sci. 2021, 22, 3817-3844. https://doi.org/10.3390/ijms22083817.

This review summarizes the studies devoted to the development of Pt(IV) prodrugs with axial ligands, that are non-steroidal anti-inflammatory drugs. Here the antitumor and anti-inflammatory chemotherapy are discussed in combine. Possible mechanisms and structure-activity relationship data are also discussed for these compounds.

  • https://doi.org/10.1155/2018/8276139

Xuejiao Li, Yahong Liu, Hongqi Tian: Current Developments in Pt(IV) Prodrugs Conjugated with Bioactive Ligands, Bioinorganic Chemistry and Applications, 2018, Vol. 2018, Article ID 8276139, https://doi.org/10.1155/2018/8276139.

In this review, the focus is again on the nature of axial ligands, which are bioactive targeting ligands, such as histone deacetylase inhibitors, p53 agonists, alkylating agents, and nonsteroidal anti-inflammatory agents. Because none of the compounds discussed in this review have been approved for clinical use, the authors believe that the data discussed will encourage studies toward clinical trials.

  • https://dx.doi.org/10.1021/acs.inorgchem.9b03683

Alfonso Annunziata, Angela Amoresano, Maria Elena Cucciolito, Roberto Esposito, Giarita Ferraro, Ilaria Iacobucci, Paola Imbimbo, Rosanna Lucignano, Massimo Melchiorre, Maria Monti, Chiara Scognamiglio, Angela Tuzi, Daria Maria Monti, Antonello Merlino, Francesco Ruffo: Pt(II) versus Pt(IV) in Carbene Glycoconjugate Antitumor Agents: Minimal Structural Variations and Great Performance Changes, Inorg. Chem. 2020, 59, 4002–4014, https://doi.org/10.1021/acs.inorgchem.9b03683.

This article presents an outstanding experimental work on the synthesis, characterization and biological properties of platinum coordination compounds (with different oxidation state of platinum) obtained with a glycoconjugate carbene ligand. Here the effects of the oxidation state of platinum and the structural features of the isostructural complexes were discussed and compared in respect to their cytotoxicity.

In the Section 1.2 of the Introduction of our article are discussed each of the three groups of ligand applied in the construction of the inner coordination sphere of the complexes of Pt(IV), namely “non-leaving” group ligands, “leaving” ligands, as well as the axial ligands. With appropriate examples from the literature, it is shown how the knowledge for the nature, the molecular structure and physicochemical properties of the ligands from the different groups could be applied in modeling the biological properties of the compounds, that are candidates of the antitumor agents. Also, with the suitably chosen examples is shown haw it is possible to design on the basis of knowledge of the mechanisms of biotransformation and possible interactions in the biological environment from chemical point of view.

As examples, we can give many other reviews and articles that analyze the structure of Pt(IV) complexes, but we have focused on the most basic ones presented by groups with many years of experience.

To the best of our knowledge, the information presented in our manuscript does not duplicate any other in terms of approach and emphasis. And most importantly, this section should be included to achieve a clear distinction between the spatial arrangement of the ligands in the inner coordination sphere of the new compounds presented in this study and that of the classical representatives.Therefore, I strongly request the reviewer to revise and rethink this part of the work. In any case, the gross elimination of this part will not be useful for the quality of the work and also for the readers.

Comment 3. I agree entirely that the complete characterization of new molecules will be vital in describing and explaining their biological activity, particularly when the structure is retained in both solution and solid state. As mentioned by the author in response to comment 4, the description of the interactions that stabilized compound 1A is only justified in the main manuscript due to the contribution to the description of ligand kinetic exchange described in solution and studied by NMR.   The rest of the crystallographic data analysis of 1B and 2A structures should be moved to supplementary information.  

Response: Perhaps the reviewer means 1B and 1C, but not 1B and 2A.

First: The structural characterization of 2A is also important because this compound is also under biological investigation. The compound is soluble in water, but no significant hydrolysis was observed during the study period (at higher concentrations suitable for an NMR experiment). This is an experimental result, but not a bad feature. If we look in the Introduction (r. 206), it is clear, that “cisplatin-induced DNA cross-linking will not be preferred for the Pt(IV) compounds [13,16,17,21] and therefore, alternative mechanisms of their behavior would be expected”.

Second: In respect to the compounds 1B and 1C, we discuss their structure in order to support the structure of 1A and explain its physicochemical properties. As written in the article, the positions of the water molecules in the in the single-crystal structure of 1A were not implicitly solved for each molecule. The advantages from the analysis of 1B and 1C in the main text can be taken from: 1) unambiguous presentation of the spatial orientation of coordination polyhedron of the complex cation 1 considering the effect of the solvent molecules (taking in to account deformations observed in the individual compounds); 2) way to explain better solubility of 1A; 3) determination of the suitable positions for H-bonding formations and etc..

 Comment 8. Considering that h2 was identified as the main product of the hydrolysis of compound 1 in physiological conditions, the authors should provide a proposal about the effect of inner sphere modifications in reactivity and its influence on antiproliferative activity. In my opinion, Pt(IV) inertness could be proposed for [Pt(IV)(taci)(I3)]+ but not for  [Pt(IV)(taci)(I)(OH)2]+; the substitution kinetics of hydroxy ligands must contribute in the biological activity observed. 

Response: Based on the classical understanding of the mechanism of cisplatin, the proposal to explain the effect of the inner sphere of h2 on its antiproliferative activity is clear – formation of fac,cis-Pt(IV)(taci)I-DNA-adducts, but unfortunately the mechanisms can be much more complex. Furthermore, the results obtained were from an NMR experiment in concentration ranges that are much higher than those of the bioassays. Although this is an important result, I am afraid that such a statement without precise mechanistic studies may be overstated. I fully support that the substitution kinetics of hydroxy ligands is one of the main reasons contributing to the observed biological activity.

Comment 11.  I suggest removing data associated with caspase 8 and including it with the results obtained from the ongoing work. Enhance the discussion to explain the results comparing with other compounds from literature.

Dear reviewer, I'm afraid you don't believe our results. This is probably why you advised us in your first review to replace Figures 3 and 8 with new ones that present more information but are uglier, even though the information you requested was contained in and could be derived from Tables S4 and 2. We are excited by these first mechanistic results and would like to report them, although studies in this direction are ongoing.

Comment 12. I agree that much of the information in the introduction could be moved to the discussion section to explain several results obtained. In response to comment 12, the authors mention that they expected that Pt(IV) compuds undergo reduction to Pt(II) species, undergo hydrolysis, and ultimately platinate DNA. However, in table 1 the author proposes hydrolysis of compund 1 to get compound  [Pt(IV)(taci)(I)(OH)2]+ without the redox transformation.

Response: There is a misunderstanding here. The repeated information from the Introduction in the response 12 of your first report concerns the known about the common mechanisms of Pt(IV) compounds accepted. Nowhere in our paper is a proven reduction of the new compounds to the corresponding Pt(II) complexes noted.  On the contrary, it was noted that within the studied period, with the help of laboratory experiments (NMR and mass spectrometry), the chemical species in solution were found to be 1 and its Pt(IV) hydrolysis products (Тable 3). Here there isn’t an error. In our manuscript it is also written that our “results supported a different mechanism …in contrast to cisplatin and its analogues” – r. 897.

 Additional reviewer questions:

  • Why do they propose hydrolysis before the redox reaction if they expected the redox process and then the hydrolysis?

Response: If we turn to the literature, it is clear that it is difficult to arrange the processes of reduction and hydrolysis one after the other. Although it is generally accepted that reduction precedes hydrolysis, from a chemical point of view it is clear that the equilibria that are established in biological environments are very complex and therefore the answer to this question is not clear cut. For our compounds, we report only the observed and studied physicochemical properties and nowhere note the preparation of Pt(II) species.

  • Could compound h2 platinate DNA without suffering the redox transformation? 

Response: The paper does not report experimental results, by which to provide a clear answer to this question, as the main purpose of this paper is to reporting the structurally novel design of an antitumor drug candidates endowed with promising pharmacological properties and to report their synthesis and structure in solid state and solution.

  • Does it influence the biological activity of the compound?

Response: The answer is the same: The paper does not report experimental results to provide a clear answer to this question, but the answer should be positive.

What can the authors say about this?

 Response: If we turn again to the literature, it is clear that the mechanistic studies that can give a correct and true answer to these questions are in the largest volume only for cisplatin. For all scientists working in this field - biologists, pharmacists, chemists, physicists, spectroscopists, etc., the answer to these questions is a huge challenge, and everyone is looking for the right answers to these questions, but the results obtained are not always unambiguous. The answers at the stage of our research would be only guesses which need not be discussed at this stage.

Last but not least, we would like to thank the reviewer for these questions. They mean that the reviewer was interested in our compounds and gave them a high rating.

22.08.2022                             Georgi Momekov and Galina Gencheva

Please see the attachments also

Round 3

Reviewer 1 Report

Dear authors, I believe that the results presented are fascinating. However, some points remain unclear to me.

-The inclusion of the IC50 results seems very important to me in the abstract of the article. Especially since regardless of whether the cells are sensitive (HL60) or resistant (HL60/Dox and HL60/CDDP), the IC50 value when exposing said cultures to compound h2 is practically the same. In my opinion, it is one of the most relevant results of the work.

I do not know if the text that the authors mention in their response to insert on line 34 seems to reflect what I say or if its statement is the most relevant result of the in vitro tests.

-Much of the information provided in the introduction could well be used to discuss in detail the results obtained, making it even more evident that the results they present cannot be described with the known mechanisms for different Pt(II) and Pt compounds. (IV).

From my point of view, making the introduction shorter and the discussion broader allows better monitoring of the results and identification of their relevance.

I leave the final decision on this point to the editor's discretion.

-Regarding the solid state section, my comment is intended to broadly discuss only structure 1A [Pt(IV)(taci)(I3)]+ and mention the generalities of structure 2A. The rest of the crystallographic structures 1B and 1C can appear in summary form in the main text and the rest of the discussion in the supplementary material. I do not consider it relevant to make the discussion of the X-ray structures of structures 1B and 1C so extensive, especially since I did not find anywhere in this section the elements that they claim to explain as a better solubility and the determination of the positions of the hydrogens to form hydrogen bonds (positions that could be changing in solution as a function of the solvent and other molecules present in there). Please, mark in the text where the authors mentioned the points with the analysis of the X-ray structures are discussed; I could not find them.

Unless you want to reorient your discussion to give the most significant importance of the manuscript to the structural characterization and the characteristics of these compounds in the solid state, I again recommend cutting out the X-ray section.

-I believe in your results, but I also think that with all the results you obtained, you can make better proposals as to how these Pt(IV) species could generate cytotoxicity in cell cultures. The literature mentions that a reduction generally occurs before hydrolysis, a fundamental step for interaction with DNA. However, you describe a Pt(IV) species that already have OH units in its coordination sphere, and you don't mention anything about it in your discussion. It seems a highly relevant fact for everything the literature says, but I don't see it reflected in the article. That is why my insistence on the proposals of the possible mechanisms.

I am not asking for hard data to be presented on what these compounds do at the molecular level. I ask that with the results of the characterization in solution, the cytotoxic effects observed, and the information provided by the literature regarding different Pt(IV) compounds, they propose what these compounds can do to kill different cell types. I imagine that for this reason, they are sending their article to Pharmaceutics and not to another journal where structural information is the most important and the cytotoxic response is just one more piece of information.

Author Response

Dear Reviewer,
Thank you for taking the time to review and to work on our manuscript. All your comments, corrections and suggestions during the three Rounds were very useful and contributed to improved the manuscript.

Response 1. The Abstract has been revised and the requested data were included.

Response 2. The Introduction was condensed:

  • The subtitles were removed;
  • The part 1 was condensed to four sentences;
  • We tried to remove the redundant information from parts 2 and 3;
  • We tried to be "bold" enough to transfer information from the introduction to the discussion.

Together with the editing of the Introduction, the List of References was reorganized and condensed.

Response 3. The main purpose of this paper is to report a structurally novel design of compounds candidates for antitumor drugs, and we wanted to present the structural characterization and biological properties of  these compounds with equal contribution.  We considered that from a chemical point of view, the structure analysis of 1B and 1C, which gives an additional support of description of the structure of 1A and for the explanation of its physicochemical properties, is self-evident. However, we have tried to improve this part by providing information in the Supplementary Information.

Response 4. We have tried to improve the MS according to Reviewer’s suggestion.

We also tried to improve the English language and style as recommended by the Reviewer using the Track Changes function.

September, 16th 2022                                         Galina Gencheva 

Round 4

Reviewer 1 Report

The authors considered most of the suggestions and made the necessary modifications. The manuscript is ready for publication.